



# Dissolved Pb and Pb isotopes in the North Atlantic from the GEOVIDE transect (GEOTRACES GA-01) and their decadal evolution

Cheryl M. Zurbrick[1], Edward A. Boyle[1], Rick Kayser[1], Matthew K. Reuer[1], Jingfeng Wu[1],
Hélène Planquette[2], Rachel Shelley[2], Julia Boutorh[2], Marie Cheize[2], Leonardo Contreira[3], Jan-Lukas Menzel Barraqueta[4], and Géraldine Sarthou[2]

[1]Earth, Atmospheric and Planetary Sciences, Massachusetts Institute of Technology, Cambridge, MA 02139, USA
[2]Laboratoire des Sciences de l'Environnement Marin (LEMAR) ,Institut Universitaire Européen de la Mer, Technopôle Brest-Iroise,13 Plouzané 29280, France
[3] Universidade Federal do Rio Grande (FURG), Institute of Oceanography, Rio Grande, Brazil
[4] GEOMAR, Helmholtz Centre for Ocean Research, Kiel, Wischhofstraße 1-3, Build. 12, D-24148 Kiel, Germany

*Correspondence to*: Edward A. Boyle (eaboyle@mit.edu)

**Abstract.**

During the 2014 GEOVIDE transect, seawater samples were collected for dissolved Pb and Pb isotope analysis. These samples provide a high resolution "snapshot" of the source regions for the present Pb distribution in the North Atlantic Ocean. Some of these stations were previously occupied for Pb from as early as 1981, and we compare the 2014 data with these older data some of which are reported here for the first time. Lead concentrations were highest in subsurface Mediterranean Water (MW) near the coast of Portugal, which agrees well with other recent

observations by the U.S. GEOTRACES program (Noble et al., 2015). The recently formed Labrador Sea Water (LSW) between Greenland and Nova Scotia is much lower in Pb concentration than the older LSW found in the Western European Basin due to decreases in Pb emissions into the atmosphere during the past 20 years. Comparison of North Atlantic data from 1989 – 2014 shows decreasing Pb concentrations consistent with decreased anthropogenic inputs, active scavenging, and advection/convection. The nearly-homogenous isotopic composition of

northern North Atlantic seawater implies that the relative proportions of U.S. and European Pb sources to the ocean have been relatively uniform during the past two decades. Using our measurements in conjunction with emissions inventories, we support the findings of previous atmospheric analyses that up to 50% of the Pb deposited to the ocean in 2014 was natural, although it remains unclear if that natural dust is from the mid- or high-latitudes.

**1 Introduction**

Humans have greatly perturbed the biogeochemical cycle of Pb, with the most dramatic changes during the 1950s – 1990s (Schwikowski et al., 2004). This resulted in large increases of Pb to not only local environments (Harris & Davidson, 2005) but to remote areas such as Greenland (Bory et al., 2014) and Antarctica (Rosman et al., 1994). Because Pb is a potent neurotoxin (ATSDR, 2007), efforts to reduce anthropogenic Pb emissions were widespread





throughout the 1980s – 1990s. Since the phasing out of leaded gasoline by most northern European and American countries and passage of other forms of clean air regulation, atmospheric Pb emissions have declined dramatically in the past 3 decades (EMEP WebDab, 2017). As a result, far less Pb has been mobilized into the atmosphere and less deposited in remote places such as the open ocean.

Pb pollution in the North Atlantic Ocean has been studied more than the other ocean basins. The United States consumed the largest quantity of leaded gasoline of any nation from 1930 – 1980, and carried by the prevailing Westerly winds (30° – 60° N), this produced the most visible oceanic contamination in the North Atlantic Ocean. Of relevance to this study (Figure 1), surface Pb concentrations ([Pb]) were measured in 1981 (TTO, Weiss et al.,

2003), 1989 (Atlantis II 123, this work; JGOFS, Martin et al., 1993), 1993 (IOC-2, Veron et al., 1999), and 1999 (Endeavor 328, Noble et al., 2015 and this work). More recent campaigns through the GEOTRACES program have occurred in 2010 (GA02, The GEOTRACES Group, 2015) and 2010/2011 (GA-03, Noble et al., 2015).

In the western North Atlantic, repeat sampling of time series locations have documented the reduction in oceanic

[Pb] and changes in sources with time. At BATS (Bermuda Atlantic Time Series) in the 1970s and 1980s, concentrations were 80 – 160 pmol kg$^{-1}$ near the surface but 25 pmol kg$^{-1}$ at depth (Boyle et al., 2014 and references therein). As Pb emissions were reduced and surface waters subducted, the elevated [Pb] could be seen as a plume in subsurface waters at increasingly deeper depths over time. At the latest occupation of BATS (2011), surface water concentrations were less than 20 pmol kg$^{-1}$ (Noble et al. 2015). Despite a dramatic reduction in [Pb], it is still

believed that a large fraction of that Pb is a result of (coal) combustion and industrial processes based on positive matrix factorisation analysis of aerosols (Shelley et al., 2017; Noble et al 2015). In the tropical Atlantic, another 2010 - 2011 study found that 50 – 70% of Pb in the surface ocean was anthropogenic in origin (Bridgestock et al., 2016), with the remaining fraction from natural North African dust.

This study evaluates current sources and relative quantities of Pb in the northern North Atlantic Ocean. We compare these findings with older seawater Pb data (some published for the first time here). Our study is strongly enhanced by the partnership of the environmental trace metal GEOTRACES program with the OVIDE program's long-term studies of physical oceanographic parameters in the Northeast Atlantic (Garcia-Ibanez et al., 2015).

**2 Methods**

**2.1 Sample Collection**

The GEOVIDE cruise track began in Lisbon, Portugal on 15 May 2014 and followed the OVIDE section from the Iberian upwelling system to the subpolar North Atlantic region up to the Greenland margin before continuing on to the Labrador Sea at the Canadian margin, finishing on 30 June 2014. One liter Nalgene HDPE sample storage

bottles were acid cleaned and stored, double-bagged as previously described (Noble et al., 2015). Trace metal clean seawater samples were collected using the French GEOTRACES clean rosette (General Oceanics Inc. Model 1018





Intelligent Rosette), equipped with new, cleaned 12L GO-FLO bottles (Cutter and Bruland, 2012). The rosette was deployed on a 6mm Kevlar cable with a dedicated custom designed clean winch. Immediately after recovery, GO-FLO bottles were individually covered at each end with plastic bags to minimize contamination. They were then transferred into a clean container (class-100) for sampling. For Stations 1, 11, 15, 17, 19, 21, 25, 26, 29, 32 samples

5  were filtered with 0.2 µm capsule filters (SARTOBRAN® 300, Sartorius). For all other stations (13, 34, 36, 38, 40, 42, 44, 49, 60, 64, 68, 69, 71, 77) seawater was filtered directly through paired filters (Pall Gelman Supor 0.45µm polystersulfone, and Millipore mixed ester cellulose MF 5 µm) mounted in Swinnex polypropylene filter holders, following the Planquette and Sherrell (2012) method. All samples were acidified back in the MIT laboratory with 2mL trace metal clean 6M HCl per liter of seawater (final pH ~2).

Previously unpublished Pb and Pb isotope data from cruises from 1989 (Atlantis II cruise 123) and 1999 (cruise Endeavor EN328) are included here for evaluation of the decadal evolution of Pb in the eastern North Atlantic. We supplement our 1989 data with two published JGOFS stations (Martin et al., 1993). Our 1989 samples were collected using "vane bulb" samplers (Boyle et al., 1986) and the 1999 samples were collected using the MITESS

mooring sampler (Bell et al., 2002). Samples were stored in acid-cleaned 250ml LPE bottles.

### 2.2 Pb Concentrations

GEOVIDE samples were analysed at least 1 month after acidification during more than 36 analytical sessions using the isotope-dilution ICP-MS method described in Lee *et al*. 2011, which includes pre-concentration on

nitrilotriacetate (NTA) resin and analysis on a quadrupole ICP-MS (Fisons PQ2+). Method details including all cleaning protocols are available in the metadata file, along with the data, in the BCO-DMO repository (see 2.4).

Briefly, triplicate subsamples (1.3mL) were spiked with a known $^{204}$Pb spike and the pH was raised to 5.3 using a trace metal clean ammonium acetate buffer, prepared at a pH of between 7.95 and 7.98. ~2400 beads of cleaned NTA Superflow resin (Qiagen Inc., Valencia, CA) were added to the mixture and equilibrated. After equilibration,

the resin was rinsed with distilled water and then Pb was eluted with a 0.1M solution of trace metal clean HNO$_3$ before analysis by ICP-MS.

On each day of sample analysis, procedural blanks were determined for 12 replicates of in-house reference seawater with negligible [Pb]. The blanks analysed concurrently with these samples ranged from 2.2 – 9.9 pmol kg$^{-1}$, averaging 4.6 ± 1.7 pmol kg$^{-1}$. Within a day, procedure blanks were very reproducible with an average standard

deviation of 0.7 pmol kg$^{-1}$, resulting in detection limits (3x the low-level standard deviation) of 2.1pmol kg$^{-1}$. Replicate analyses of three different large-volume seawater samples (one with ~11 pmol kg$^{-1}$, another with ~24 pmol kg$^{-1}$, and a third with ~38 pmol kg$^{-1}$) indicated that the precision of the analysis is 4% or 1.6 pmol kg$^{-1}$, whichever is larger. Triplicate analyses of an international reference standard SAFe D2 were 27.2 ± 1.7 pmol kg$^{-1}$.





Pb concentration analysis for 1989 samples (Atlantis II 123) was achieved by $^{204}$Pb isotope dilution with Mg(OH)$_2$ coprecipitation followed by VG PQ2+ quadrupole ICPMS (Wu & Boyle, 1997) (analysed in 1996) and 1999 (Endeavor 328) stations 4, 5, 7, 9, 10 and 11 (analysed between 1999-2003). Endeavor 328 stations 2, 3, 8, and 10 were determined using NTA-extraction ID ICPMS (Lee et al., 2011) (determined in 2010). Long-term quality
control seawater samples were included in each run, and overlapped with new QC samples when the previous samples were depleted. Endeavor 328 station 10 was determined twice by two analysts 8 years apart (in 2002 by Mg(OH)$_2$ coprecipitation ID-ICPMS, and in 2010 by NTA extraction ID-ICPMS). A regression of the 2010 vs 2002 data forced through the origin had a slope of 0.945. We suggest that this small offset provides a reasonable estimate of our inter-decadal analytical reproducibility. It also demonstrates that Pb is not continuously leached from well-
cleaned LPE bottles during decadal-scale storage.

### 2.3 Stable Pb Isotopes

GEOVIDE samples were analysed during 11 mass spectrometry sessions by the method of Reuer et al. (2003) as modified by Boyle et al. (2012). In brief, ~500mL of seawater was pre-concentrated using a low-blank double
magnesium hydroxide co-precipitation, induced by minimal addition of high-purity ammonia solution and mixing (typically 8µL ammonia per 1mL seawater sample). The precipitate was dissolved in a minimal amount of high-purity 6M HCl before undergoing another ammonia addition and second Mg(OH)$_2$ coprecipitation. The final precipitate was dissolved in ~1mL of high purity 1.1M HBr the day of purification by anion exchange chromatography (Eichrom AG1x8). Samples were dried and stored in PTFE vials until isotope ratio analysis on a
GV/Micromass IsoProbe multicollector ICPMS using an APEX/SPIRO desolvator. Just before analysis, samples were dissolved for several minutes in 10µl concentrated ultrapure HNO$_3$ followed by addition of 400µL of ultrapure water and spiked with an appropriate amount of Tl for mass fractionation correction. IsoProbe multicollector ICPMS Faraday cups were used to collect on $^{202}$Hg, $^{203}$Tl, $^{205}$Tl, $^{206}$Pb, $^{207}$Pb, and $^{208}$Pb. An Isotopx Daly detector with a WARP filter was used to collect on $^{204}$Pb+$^{204}$Hg. Because the deadtime of the Daly detector varied from day to day,
we calibrated deadtime on each day by running a standard with known $^{206}$Pb/$^{204}$Pb at a high 204 count rate. The counter efficiency drifts during the course of a day, so we established that drift by running a standard with known $^{206}$Pb/$^{204}$Pb (and a 204 count rate comparable to the samples) every five samples. Tailing from one Faraday cup to the next was corrected by the $^{209}$Bi half-mass method as described by Thirlwall (2000).

On each analytical date, we calibrated the instrument by running NBS981 and normalized measured sample isotope ratios to our measured raw NBS981 isotope ratios to those established by Baker et al. (2004). Using this method for 22 determinations of an in-house Pb isotope standard solution shows that for samples near the upper range of the Pb signals shown for samples (~1V), $^{206}$Pb/$^{207}$Pb and $^{208}$Pb/$^{207}$Pb were reproduced to ~200ppm. Low-level samples will be worse than that, but generally better than 1000ppm in this data set. Because of the drift uncertainty in the Daly
detector, $^{206}$Pb/$^{204}$Pb for samples in the mid-to-upper range of sample concentrations will be reproducible at best to ~500ppm.



We have intercalibrated Pb isotope analyses with two labs as reported in Boyle et al. (2012). The outcome of that intercalibration suggests that the accuracy of our measurements approaches the internal analytical reproducibility we note above.

Pb isotope precision for the complete analytical procedure can be assessed by duplicate measurements of samples. In most cases, the replicated samples were chosen because they fell off of the trend of adjacent samples. That could be due either to contamination of the subsample used for the analysis, or to the contamination of the sample in its primary sample bottle. As shown in Figure S2, the replicate analysis usually agreed within better than 1000ppm for $^{206}Pb/^{207}Pb$ and $^{208}Pb/^{207}Pb$, and 5000ppm for $^{206}Pb/^{204}Pb$. We suggest that these provide a reasonable upper limit for

the replicability of our isotope measurements.

Pb isotope data from the 1999 samples were obtained by IsoProbe Multicollector ICPMS after $Mg(OH)_2$ preconcentration and anion exchange purification as described by Reuer et al. (2003). As for the GEOVIDE samples, the mass spectrometer was calibrated using NBS981.

### 2.4 Data Management

All [Pb] and isotope data related to the GEOVIDE data set in this manuscript have been submitted to BCO-DMO and will be available at (http://www.bco-dmo.org/dataset/651880/data and http://www.bco-dmo.org/dataset/652127/data) and from the 2017 BODC International GEOTRACES Intermediate Data Product v2

(The GEOTRACES Group, 2015). All other data is available in table 1.

## 3. Results and Discussion

### 3.1 Outliers

In this data set, we did not encounter any samples that did not yield acceptably reproducible results upon repeated analysis, so we believe that the data truly represents the concentration of Pb in the sample collection bottle.

However, there were a few samples with elevated Pb based on adjacent samples and for which no obvious hydrographic argument could be made for the anomaly. We observed that the samples taken from GOFlo in rosette position 1 (usually the near-bottom sample) were always higher in [Pb] than the samples taken immediately above that, and that the excess decreased as the cruise proceeded (Figure S1). The Pb isotope ratio of these samples were higher than the comparison bottles as well. At two stations where our near-bottom sample was taken from rosette

position 2 rather than 1, there was no Pb excess over the samples immediately above. We believe that this evidence points to GoFLO bottle-induced contamination that was being slowly washed out during the cruise, but never completely. A similar pattern was observed for the samples taken from rosette positions 5, 20 and 21 when compared to the depth-interpolated [Pb] from the samples immediately above and below. We do not believe that these samples should be trusted as reflecting true ocean [Pb], so all of the samples from these GOFlos are excluded

in our discussion of this work, although they are included and flagged as unreliable within the data repositories.



In addition, we observed high [Pb] in most of the samples from Station 1 and very scattered Pb isotope ratios. The majority of these concentrations were far in excess of those values observed at nearby Station 11, and also the nearby USGT10-01 (Noble et al., 2015). Discussion among other cruise participants revealed similarly anomalous

data for other trace metals (e.g., Hg species; personal communication with L.-E. Heimburger). After discussion at the 2016 GEOVIDE post-cruise workshop, we came to the conclusion that this is evidence of GoFlo bottles not having sufficient time to "clean up" prior to use, and that most or all bottles from Station 1 were contaminated. Station 1 data is not discussed in this work, but as with the suspicious GOFlos throughout the cruise, the Station 1 data are included and flagged as unreliable in the data repositories.

**3.2 Near-surface Ocean**

Near-surface waters (11 – 20 m) displayed a moderate range in [Pb] across the transect (Figure 2). The highest concentration was located near the Portugal coast (30 pmol kg$^{-1}$). Lead concentrations decreased three-fold with distance from the coast, down to 11.5 pmol kg$^{-1}$, in the core of the far arm of the North Atlantic Current. An excellent pictorial representation of the relevant water masses discussed here can be found in Garcia-Ibanez et al.

(2017). Near-surface concentrations were higher in the Iceland Basin and Irminger Sea (Stations 21 – 60; 18.8 – 23.5 pmol kg$^{-1}$), and in Station 64, just past the tip of Greenland. The remainder of the Labrador Sea (Stations 68 - 77) had lower [Pb] (12.1 – 16.2 pmol kg$^{-1}$).

The pattern of decreasing [Pb] over the Iberian Abyssal Plain (Stations 11 – 19) correlates strongly with increasing

distance from shore (Pearson's correlation, r = -0.989, p < 0.001). This finding agrees well with atmospheric deposition models that show higher dust inputs closer to the African continent (Schepanski et al., 2009). Stations located north of 55° in the meandering NAC have higher concentrations than those in the Western European Basin. Although dust deposition to the North Atlantic Ocean is typically associated with North African dust from the Saharan Desert, Prospero et al. (2012) and Bullard et al. (2016) found that high latitude dust emissions, specifically

volcanic-based soils from Iceland, could be substantial enough to impact oceanic Fe cycling; therefore we suggest that the elevated Pb in the near-surface waters of the Icelandic Basin and Irminger Sea may possibly be dust-derived. In the GEOVIDE shipboard aerosol data (Shelley et al., 2017), Pb concentrations were high in Iceland Basin but low in the Irminger Sea. However, as Pb has a residence time of ~1 year in this region, seasonal changes in the flux could account for this discrepancy. As the North Atlantic Current becomes the Irminger Current near Greenland and

joins with the East Greenland Current, they wrap around the southern tip of Greenland and flow toward the Arctic Circle. This entrains Pb into the northeast part of the Labrador Sea, whereas the remainder of the Labrador Sea is influenced by the Labrador Current, returning from the Arctic, which has low [Pb].

Despite the variations in [Pb] across the Atlantic Ocean, Pb isotope ratios were relatively homogenous throughout

the section, and largely decoupled from the [Pb] patterns (Figures 3, 4). $^{206}$Pb/$^{207}$Pb isotope ratios varied from 1.178 – 1.186, with the majority of samples analysed being 1.180 – 1.183. $^{208}$Pb/$^{206}$Pb and $^{206}$Pb/$^{204}$Pb isotope ratios



showed similar minimal variability. No trend in isotope ratios was observed in the Iberian Abyssal Plain extending away from the coast. The low variability of isotope ratios indicates that the majority of Pb in the Northern Atlantic Ocean is well mixed in the atmosphere prior to deposition. The relatively low [Pb] and similar isotope ratios contrast sharply with surface water measurements from the previous century (Figures 5, 6). During the 1970s – early 1990s,

the predominant source of Pb to the North Atlantic was U.S. leaded gasoline (Weiss et al, 2003; Martin et al., 1993; Veron et al., 1999), which was reflected in the high $^{206}Pb/^{207}Pb$ isotope ratios (~1.20).

The mixed layer [Pb] nearest the Iberian Peninsula (30 pmol kg$^{-1}$ ) are lower than those measured by the 2010 US GEOTRACES expedition (42 pmol kg$^{-1}$ ), which we attribute to the much closer proximity of the US GEOTRACES

station to the coastline (50 km) than GEOVIDE station 11 (280 km). As mentioned previously, [Pb] at GEOVIDE stations 11 – 19 have a strong inverse correlation to distance from shore, and adding USGT10-01 (GA-03) maintains this high correlation (Pearson correlation, r = -0.990, p < 0.001). Isotopically, the USGT10-01 near-surface waters are similar to GEOVIDE station 11, indicating similar Pb sources in recent years.

### 3.3 Iberian Abyssal Plain (S11 – S19) and Western European basin (S21 – S29)

Overall, [Pb] measured from this cruise were highest in the subsurface waters of the Iberian Abyssal Plain (Station 13). The core of the elevated concentrations (~61 pmol kg$^{-1}$, Station 13) was at ~1200m deep and several hundred kilometres from the coast. This subsurface plume of Pb (concentrations of 40 – 50 pmol kg$^{-1}$) was dispersed throughout the Iberian Abyssal Plain at depths of 700 – 2000m. The Pb plume was less pronounced in the rest of the Western European basin with concentrations of 30 – 40 pmol kg$^{-1}$. Extended Optimum MultiParameter (eOMP)

water mass analysis shows that this elevated [Pb] coincides with Mediterranean Water (MW) from 700 – 1500m and Labrador Sea Water (LSW) from 1500 – 2000m (Garcia-Ibanez et al., 2017). Our finding is in good agreement with [Pb] in MW measured in 2010-2011 by Noble et al. (2015) and highlights the high [Pb] previously found in the Mediterranean Sea (Moos and Boyle, in preparation). In the lower portion of the plume, the LSW in the Iberian Abyssal Plain and Western European basin is among the oldest water sampled during this expedition. According to

CFC-11 data, LSW in this region has a combined age (subduction plus admixed relic age) of ~25 years (Fine, 2011). That age and the elevated [Pb] observed are consistent with the atmospheric Pb emissions by North America and Europe in the 1980s. The isotope ratios further support this finding, as the ocean interior has similar isotope ratios throughout ($^{206}Pb/^{207}Pb$ = 1.1832 +/- 0.0025, 1σ; $^{208}Pb/^{206}Pb$ = 2.4525 +/- 0.0024, 1σ), but are distinguishably more like US aerosols from the early 1990s (Bollhofer & Rosman, 2001) at the core of the Pb maximum (Station 13,

$^{206}Pb/^{207}Pb$ = 1.1894; $^{208}Pb/^{206}Pb$ = 2.4544; Figure 5).

The offshore profiles (Stations 13 – 29) showed consistent decreases in [Pb] in the MW and LSW from 1989 (JGOFS S19) and 1999 (Endeavor 328 S15, 17, 21) to 2014 (Martin et al., 1993; this work). In the 10 – 15 years between sampling events the Pb maxima advected into the ocean interior as the more shallow waters were ventilated

with lower-Pb surface waters, a trend also seen in the western North Atlantic near Bermuda (Boyle et al., 2012).





Below the broad subsurface plume, water mass analysis indicates depths greater than 2500m are predominantly Northeast Atlantic Deep Water (NEADW) that contains a major component of Antarctic Bottom Water (AABW) as evidenced by high silica concentrations (Garcia-Ibanez et al., 2017). In the NEADW, [Pb] were 10 – 20 pmol kg$^{-1}$, and similar to previous sampling campaigns nearby in 1989 and 1999 (Figure 5). Isotope ratios ($^{206}$Pb/$^{207}$Pb = 1.1827

± 0.0013; $^{208}$Pb/$^{206}$Pb = 2.4511 ± 0.0013) were also similar across the 25 years in the Western European Basin (Figure 6). This makes sense because the estimated age of NEADW is several hundred years (Matsumoto, 2007).

Below 1000m, the [Pb] at Stations 11 and 13 were very similar to the 2010 [Pb] measured on GA-03 (USGT10-01; Figure 5), but the isotope ratios are dissimilar (Figure 6). Conversely, the upper 1000m of the water column had

different [Pb] but similar isotope ratios. In the upper ocean, this discrepancy can be related to the distance of the stations from shore, as calculated in section 3.2, with greater Pb inputs and therefore greater concentrations at stations closer to shore. In the deep ocean, the contrast in isotope ratios between the more coastal GA03 station and offshore GEOVIDE station, only 4 years apart, imply that the two cruises were sampling in different water masses, either because they sampled ~250 km apart or due to a boundary shift that occurred in the 4 years between sampling

events.

### 3.4 Icelandic Basin (S32 – S36) and Reykjanes Ridge (S38)

In the Icelandic Basin and above the Reykjanes Ridge, [Pb] throughout the water column are similar to those found in the Western European basin, with a subsurface [Pb] maxima (~30 pmol kg$^{-1}$) in the core of LSW. In the deepest samples (2500 – 3000m), [Pb] (5 – 10 pmol kg$^{-1}$) are lower than the NEADW observed in the Iberian Abyssal Plain

and Western European basin, and the $^{206}$Pb/$^{207}$Pb isotope ratios are slightly lower ($^{206}$Pb/$^{207}$Pb = 1.1812 ± 0.0005) than the overlying water 800 – 2000m ($^{206}$Pb/$^{207}$Pb = 1.1845 ± 0.0014). Water mass analysis indicates very little NEADW was present in the Icelandic Basin, and the deeper samples were strongly influenced by Iceland-Scotland Overflow Water (ISOW), particularly at Stations 32 – 36 (Garcia-Ibanez et al., 2017). The 1993 IOC-2 survey by Veron et al (1999) found ISOW ($^{206}$Pb/$^{207}$Pb = 1.173 – 1.176) isotopically distinct from LSW ($^{206}$Pb/$^{207}$Pb = 1.190 –

1.20) and that ISOW reflected atmospheric emissions from Europe at that time. The differences in Pb isotopes (and two to three-fold reduction in concentrations) between sampling campaigns highlights the young age of ISOW, which reflected large source changes over a 21 year time period (Figures 5, 6).

In addition, we note that the present-day Norwegian Sea waters must have low [Pb], and that their Pb isotope ratios

reflect a greater contribution from European sources than North American sources. ISOW is formed as a mixture of LSW and Norwegian Sea water that overflows the Iceland-Scotland sills. Because LSW has higher [Pb] and heavier $^{206}$Pb/$^{207}$Pb isotope ratios than ISOW, Norwegian Sea water must have a lower $^{206}$Pb/$^{207}$Pb isotope ratio and much lower [Pb]. Using our Pb data for Station 32 and the eOMP analysis that the deepest samples are 100% ISOW and ~20% LSW (Garcia-Ibanez et al., 2017), we back-calculate a Norwegian Sea water that is ~7 pmol kg$^{-1}$ and

$^{206}$Pb/$^{207}$Pb ~1.180. The relatively lower $^{206}$Pb/$^{207}$Pb isotope ratios of the Norwegian Sea are consistent with what





Veron et al. (1999) observed in 1993 (1.169), and are indicative of atmospheric Pb from a more European provenance than North American one (Figure 7).

### 3.5 Irminger Sea (S40 – 60)

In the Irminger Sea, a broad Pb maxima with little concentration variability was observed between the near surface and 1800m. The diffuse elevation in [Pb] throughout the upper 1800m is attributed to both Irminger Subpolar Mode Water (0 – 1000m) and LSW (500 – 2500m) (Garcia-Ibanez et al., 2017). As in the Icelandic Basin, ISOW is observed in the Irminger Sea deep water, but in a lower proportion (40 – 60%) than in the Icelandic Basin (80 -

100%). At Stations 42 and 44 ISOW is distinguished by its low [Pb] (5 – 8 pmol kg$^{-1}$) and a low $^{206}$Pb/$^{207}$Pb ratio (1.1798). Further north in the Irminger Sea along the Greenland continental slope, the near-bottom samples at Stations 49 and 60 are Denmark Straight Overflow Water (DSOW). The DSOW has slightly higher [Pb] (10 – 18 pmol kg$^{-1}$) and a higher $^{206}$Pb/$^{207}$Pb ratio (1.1854) than ISOW, consistent with the 1993 data of Veron et al. (1999; $^{206}$Pb/$^{207}$Pb = 1.179 – 1.182). DSOW is a mix of the Nordic Sea waters overflowing the Greenland-Iceland sill and

mixing with LSW; DSOW is also reported to have inputs from dense Greenland shelf water and cascading Polar Intermediate Water (Garcia-Ibanez et al., 2015; this study). The resulting DSOW isotope composition is very similar to LSW, which could indicate shelf water has very little Pb and so its signal is dominated by the LSW signal, although we cannot rule out the possibility that the shelf water entrained Pb with a similar isotope composition to LSW.

The Irminger Sea was previously sampled for Pb during the 1993 IOC-2 expedition (Veron et al., 1999) and the 2010 GA02 expedition near GEOVIDE Stations 42 and 44 (analyses by Middag and Bruland as reported by The GEOTRACES Group, 2015). There is a strong decrease at all depths from 1993 to 2010, and a surprisingly large decrease between 2010 and 2014. We suspect that difference between 2010 and 2014 could also be a result of the

2012 deep winter convection event (~1200m) as reported by Fröb et al. (2016). The $^{206}$Pb/$^{207}$Pb values between 1993 and 2014 do not appear to have changed significantly (perhaps in view of limited 1993 water column coverage).

### 3.6 Labrador Sea (S64 – 77)

In the Labrador Sea, the Pb maximum coincides with LSW (0 – 2500m) and is very broad. Similar concentrations (~25 pmol kg$^{-1}$) are found from 100m to nearly 2000m. At depths greater than 2000m, the [Pb] decreases to ~8 pmol

30    kg$^{-1}$ and water mass analysis indicates this is primarily ISOW. Throughout the entire Labrador Sea water column Pb isotope ratios are homogenous, in contrast to the Icelandic and Irminger basins, which are isotopically distinctive from overlying LSW. The similarity of the Pb throughout the Labrador Sea can be attributed to deep winter convection that annually varies from 1000m – 2000m deep (Lazier et al., 2002 Deep Sea Research I; Lilly et al., 1999 J Phys Oceanog; Vage et al., 2009). Hydrographic observations and Argo floats indicate winter 2014

convection was ~1700m deep (Kieke & Yashayaev, 2015). Fine (2011) assigns a combined age of 17 – 19 years to





these waters. The similar Pb profiles throughout the entire water column indicate there were minimal changes in Pb sources to the LSW over the 2 decades preceding sampling, and the isotopically indistinguishable ISOW suggests it is also relatively well-mixed with LSW in this basin.

The Labrador Sea also confirms the continued changes to oceanic Pb since the phase-out of leaded gasoline usage by North America and Europe. Pb concentrations in the upper 2000m of the water column were three to four times lower in 2014 and in 2010 than those measured in 1993 (2010 analyses by Middag and Bruland as reported by The GEOTRACES Group, 2015; Veron et al., 1999) (Figure 5). Surface water Pb isotope ratios in 2014 were also much lower ($^{206}Pb/^{207}Pb$ = 1.186) than during the early 1990s ($^{206}Pb/^{207}Pb$ = 1.209) (Figure 6), in agreement with the rest

of the North Atlantic Ocean surface Pb changes.

### 3.7 Sources of Pb in 1999 and 2014

Overall, Pb isotope ratios throughout the GEOVIDE expedition were nearly homogenous, in both the upper and

15 deep ocean, and in the eastern and western basins. This finding is highly similar to that of Noble et al. from the US GEOTRACES expeditions in the mid-Atlantic in 2010 and 2011, but differs from the expeditions of the 1980s and 1990s when Pb isotope ratios ranged much more broadly ($^{206}Pb/^{207}Pb$ = 1.165 – 1.201) (Veron et al, 1999). The near uniformity of Pb isotope ratios in 2014 throughout the water column shows that the relative proportions of isotopically distinct sources of Pb have been similar for the ~20 years preceding sampling.

Atmospheric deposition is the main source of Pb to the ocean, with surface waters reflecting the most recent inputs. Trace metal enrichment factors of dry aerosols and wet deposition were collected during the GEOVIDE cruise (Shelley et al., 2017). Results for Pb enrichment were moderate (>10), suggesting much of the atmospheric Pb was derived from anthropogenic sources. Using positive matrix factorization of the aerosol concentration data, Shelley et

al. estimated that ~60% of the Pb was from a mineral dust source and only 40% was of anthropogenic origin. This finding parallels the 2010 study of Pb in the tropical North Atlantic by Bridgestock et al (2016). That group determined 30 – 50% of the surface water Pb was of natural mineral dust from the North African dust plume. A triple isotope plot of surface waters from this cruise (Figure 7a) shows visually good agreement with Shelley et al.'s estimate that ~half of the Pb in the surface waters is of natural origin, assuming that the Icelandic dust has a similar

isotopic composition as pre-anthropogenic North African dust.

The mixed sources of Pb throughout the ocean interior are similarly clear. In Figure 7b, seawater was compared to the possible sources: Pre-Holocene sediments, atmospheric Pb measured in Europe and North America, and North African dust. The GEOVIDE Pb samples cluster tightly together and fall along the mixing line between natural Pb

and modern anthropogenic sources (i.e., industrial emissions from North America and Europe). The Pb signatures of European and North American atmospheric samples are difficult to differentiate using Pb isotopes alone, so Pb emissions estimates were evaluated using the EMEP (European Monitoring Evaluation Program) database.



Atmospheric Pb emissions for European countries along with the USA and Canada were evaluated from 1990 – 2014 (Figure S3). Cumulative atmospheric Pb emissions have reduced by a factor of 10 in Europe and by a factor of 5 in North America over that time period. The ratio of Pb emissions from USA and Canada vs European sources was 1:7 in 1990, but that ratio steadily increased to 1:3 by 1999 and has remained about the same since then, due to the much larger reductions in emissions by Europe (following upon early U.S. emission reductions). The continuity in ratio of Pb sources for the ~15 years preceding sampling and the homogenous Pb isotope ratios found throughout the North Atlantic Ocean indicate the main anthropogenic sources were North American emissions, followed by European emissions.

A triple isotope plot ($^{208}$Pb/$^{206}$Pb vs $^{206}$Pb/$^{207}$Pb, Figure 8) shows that there have been spatiotemporal Pb source changes between the 1999 EN328 cruise and regions of the 2014 GEOVIDE cruise. Most of the 1999 data (except for the oldest deep waters) fall on the lower branch of the European – U.S. mixing trend (black circles). The GEOVIDE data from stations 11-26 at all depths and the >800m samples from stations 29-77 fall on an intermediate trend, while the <800m samples from GEOVIDE stations 29-77 fall on the high side of the trend. We do not have enough source isotope information to explain these changes, but they clearly indicate spatiotemporal evolution of the evolving anthropogenic Pb transient in the northern North Atlantic Ocean.

### 3.8 Evolution of Pb and Pb isotopes in the Eastern Atlantic Water Column, 1989-2014

Data for [Pb] from the 1989 (Atlantis II 123), 1999 (Endeavor EN328), and 2010-2014 (GA-03, GEOVIDE) cruises, and Pb isotopes from 1999 and 2010-2014, are plotted as North-South sections in Figures 9 and 10. It is evident that Pb is strongly decreasing in the upper ocean during this period, a fact that can be attributed to the phasing out of tetraethyl Pb gasoline in North America and Europe. All three periods show a Pb maximum in the deep thermocline, and this maximum deepens from decade to decade, as it has also done in the western North Atlantic water column near Bermuda (Boyle et al., 2014; Noble et al., 2015). As Noble et al. demonstrated for the 2010/2011 GA-03 trans-North Atlantic section, this maximum is located in waters with SF$_6$ ventilation dates from the 1970's, when leaded gasoline Pb utilization was at its maximum. A similar result is seen for the 1989 data based on $^3$He-$^3$H dating (Jenkins, 1987). Hence the location of the maximum is dominantly a reflection of Pb emissions at the ventilation age of the water rather than an association with a particular water mass. When considered in this light – as a snapshot of an evolving three-dimensional transient tracer experiment – some of the features in these sections require an interpretation that differs substantially from that usually placed on quasi-steady-state tracers such as salinity, oxygen, and nutrients. For example the [Pb] maximum seen at ~25°N is not the source of a northward-spreading plume, it is the southern extent of high-[Pb] waters that were subducted into the thermocline in the 1970's and advected southwesterly by the dynamics of the ventilated thermocline (Luyten et al., 1983). In addition to the general ventilation of the North Atlantic water column, some [Pb] features are due to specific hydrographic features. The 1999 [Pb] maximum near 1000m was enhanced by a strong "Meddy" (a coherent mesoscale feature created by pulses of dense salty water from out of the Mediterranean Sea (Armi et al. 1989), as demonstrated by the salinity





data from that profile (Figure 11). It is also evident that the ~1800m Labrador Sea water has had consistently higher Pb than the more dense Greenland-Scotland overflow water.

It is likely that the evident decline in the Pb inventory of the eastern North Atlantic is decreasing not only because of advective/diffusive spreading of the water out of the basin, but also because of scavenging. Radiochemical studies (Bacon et al., 1976;) have shown that deep water column $^{210}$Pb activities are lower than $^{226}$Ra activities signifying removal of $^{210}$Pb from the deep water column. Some of this scavenging is due to sinking particles but in near bottom waters, "boundary scavenging" accounts for a higher fraction (Bacon, 1988).

The evolution of the Pb isotope data between 1999 and 2010-2014 is striking in that the deepest waters in the tropical Eastern Atlantic are significantly changed between these periods. Near the surface, recent changes are mainly due to a greater reduction of the relative North American high $^{206}$Pb/$^{207}$Pb sources relative to the European low $^{206}$Pb/$^{207}$Pb sources. But in the deepwater, this change probably represents the "conveyor belt" motion of deep high $^{206}$Pb/$^{207}$Pb introduced from the surface in the early 1900's being replaced by lower $^{206}$Pb/$^{207}$Pb from the 1920's and later (as seen in historical Pb isotope ratios in Bermuda corals, Kelly et al., 2009).

## 4. Conclusions

In the past 30 years, massive reductions in Pb emissions to the environment have been evidenced by sampling campaigns in the North Atlantic Ocean. Evolution of [Pb] and Pb isotope ratios will continue as human-derived emissions continually decline, Pb is naturally scavenged from the water column, and the oceanic "conveyor belt" continue to mix deep waters. Like Bridgestock et al. (2016) found in the tropical Atlantic, we see evidence of a natural Pb source to the northern North Atlantic which was previously obscured in the 1980s and 1990s by enormous anthropogenic inputs. Aerosol samples collected concurrently with our seawater samples support our determination that Pb in the surface waters is partially of natural origin (Shelley et al., 2017), and work by Prospero et al (2012) introduces the possibility that much of the dust in the Irminger Sea and Icelandic Basins is actually from high latitude sources such at Icelandic dust. Future work to better constrain end-members could validate this hypothesis.

## 5. Acknowledgements

The GEOVIDE project was funded by the French National Research Agency (ANR-13-BS06-0014, ANR-12-PDOC-0025-01), the French National Center for Scientific Research (CNRS-LEFE-CYBER), the LabexMER (ANR-10-LABX-19), and Ifremer. We also thank the shipboard technical team: Pierre Branellec, Floriane Desprez de Gésincourt, Michel Hamon, Catherine Kermabon, Philippe Le Bot, Stéphane Leizour, Olivier Ménage, Fabien Pérault, and Emmanuel de Saint-Léger. GEOVIDE nutrient data was obtained by Manon Le Goff, Emilie Grossteffan, Morgane Gallinari, and Paul Tréguer. We also thank the officers and crews of the RV Atlantis II (1989)



and RV Endeavor (1999) for their efforts on our behalf. Our GEOVIDE sample analyses were funded by the US National Science Foundation by grant OCE-1357224.





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



**Table 1: Data from 1989 Atlantis II 123 and 1999 EN328 cruises.**

| Depth | Pb | T | S |
|---|---|---|---|
| m | pmol/kg | deg C | pss |
| *Atlantis II cruise 123, Station 4, 22°N 36°E, October 15 1989* | | | |
| 1 | 44 | 27.40 | |
| 19 | 62 | 27.20 | 36.521 |
| 39 | 75 | 27.20 | 36.792 |
| 58 | 58 | 24.46 | 37.097 |
| 77 | 83 | 21.38 | 37.150 |
| 97 | 91 | 21.14 | 37.225 |
| 116 | 77 | 21.30 | 37.271 |
| 135 | 108 | 21.29 | 37.269 |
| 154 | 96 | 21.00 | 37.190 |
| 174 | 104 | 20.12 | 37.085 |
| 212 | 106 | 19.75 | 37.020 |
| 232 | 121 | 20.21 | 37.017 |
| 251 | 139 | 19.07 | 36.779 |
| 270 | 117 | 17.38 | 36.457 |
| 361 | 139 | 14.54 | 35.971 |
| 425 | 135 | 13.39 | 35.824 |
| 477 | 129 | 12.37 | 35.685 |
| 574 | 139 | 11.07 | 35.518 |
| 594 | 140 | 10.78 | 35.483 |
| 622 | 132 | 10.37 | 35.434 |
| 815 | 93 | 7.68 | 35.028 |
| 844 | 85 | | 35.004 |
| 872 | 88 | | 34.981 |
| *Atlantis II cruise 123, Station 5, 26.33°N 33.67°E, October 16 1989* | | | |
| 1 | 47 | 27.20 | 37.389 |
| 19 | 74 | 27.20 | 37.384 |
| 39 | 74 | 27.06 | 37.364 |
| 77 | 95 | 21.53 | 37.336 |
| 97 | 112 | 21.35 | 37.282 |
| 116 | 111 | 21.37 | 37.262 |
| 135 | 103 | 21.21 | 37.230 |
| 154 | 99 | 20.95 | 37.173 |
| 174 | 111 | 20.46 | 37.065 |
| 193 | 106 | 19.33 | 36.863 |
| 212 | 114 | 18.70 | 36.717 |
| 359 | 158 | 14.98 | 36.047 |
| 476 | 156 | 13.19 | 35.790 |
| 575 | 156 | 11.78 | 35.606 |



| | | | |
|---|---|---|---|
| 595 | 162 | 11.56 | 35.580 |
| 624 | 132 | 11.26 | 35.541 |
| 879 | 101 | | 35.173 |
| 1018 | 83 | | 35.092 |
| 1244 | 89 | | 35.063 |
| 1278 | 95 | | 35.069 |

*Atlantis II cruise 123, Station 7, 31°N 31°E, October 20 1989*

| | | | |
|---|---|---|---|
| 1 | 95 | 21.10 | |
| 19 | 97 | | |
| 39 | 100 | | |
| 58 | 89 | 22.59 | 35.334 |
| 77 | 94 | 19.33 | 36.795 |
| 97 | 85 | 18.84 | 36.697 |
| 116 | 94 | 18.61 | 36.655 |
| 154 | 104 | 18.09 | 36.553 |
| 174 | 100 | 17.71 | 36.488 |
| 193 | 104 | 17.33 | 36.429 |
| 212 | 117 | 16.97 | 36.363 |
| 232 | 119 | 16.58 | 36.293 |
| 366 | 134 | 14.26 | 35.928 |
| 405 | 130 | 13.74 | 35.862 |
| 484 | 135 | 12.74 | 35.730 |
| 582 | 136 | 11.67 | 35.592 |
| 601 | 136 | 11.53 | 35.575 |
| 630 | 145 | 11.31 | 35.547 |
| 826 | 100 | 9.27 | 35.404 |
| 883 | 92 | | 35.331 |
| 1244 | 75 | | 35.328 |
| 1312 | 71 | 6.44 | 35.303 |

*Atlantis II cruise 123, Station 9, 35°N 29°E, October 22 1989*

| | | | |
|---|---|---|---|
| 1 | 94 | 22.50 | 36.480 |
| 19 | 90 | 22.50 | 36.480 |
| 58 | 86 | 22.20 | 36.473 |
| 97 | 87 | 16.88 | 36.186 |
| 194 | 99 | 14.77 | 35.947 |
| 253 | 115 | 14.33 | 35.835 |
| 272 | 106 | 13.97 | 35.793 |
| 389 | 116 | 13.19 | 35.665 |
| 409 | 110 | 13.09 | 35.639 |
| 488 | 105 | 11.97 | 35.604 |
| 587 | 123 | 11.52 | 35.483 |
| 607 | 121 | 11.28 | 35.480 |
| 836 | 104 | 10.43 | 35.551 |





| 865 | 99 | 10.00 | 35.544 |
| 1294 | 88 | | 35.257 |
| 1328 | 91 | 10.67 | 35.225 |

====

| Depth m | Pb pmol/kg | Pb206/Pb207 | Pb208/Pb207 | T deg C | S permil |
|---|---|---|---|---|---|
| *Endeavor cruise 328, Station 2, 26.5°N 38.5°E, Sept. 1 1999* | | | | | |
| 0.5 | 36.9 | | | 26.600 | 37.590 |
| 48 | 39.1 | | | 26.191 | 37.573 |
| 146 | 37.9 | | | 19.896 | 36.832 |
| 196 | 41.0 | | | 18.237 | 36.552 |
| 293 | 48.3 | | | 16.741 | 36.318 |
| 441 | 60.3 | | | 14.536 | 35.974 |
| 589 | 80.3 | | | 12.503 | 35.689 |
| 687 | 89.9 | | | 10.994 | 35.505 |
| 785 | 79.4 | | | 9.246 | 35.326 |
| 931 | 65.2 | | | 7.772 | 35.195 |
| 1076 | 51.1 | | | 6.626 | 35.142 |
| 1273 | 44.6 | | | 5.820 | 35.150 |
| *Endeavor cruise 328,Station 3, 24°N 37.5° E, Sept. 2 1999* | | | | | |
| 0.5 | 25.5 | | | 21.800 | 36.120 |
| 49 | 26.9 | | | 25.947 | 37.536 |
| 98 | 30.9 | | | 22.853 | 37.385 |
| 147 | 33.2 | | | 21.215 | 37.161 |
| 194 | 36.6 | | | 19.224 | 36.785 |
| 290 | 45.9 | | | 17.000 | 36.369 |
| 429 | 61.4 | | | 14.575 | 35.979 |
| 569 | 83.2 | | | 11.940 | 35.607 |
| 653 | 83.1 | | | 10.978 | 35.495 |
| 744 | 81.1 | | | 9.455 | 35.343 |
| 883 | 63.9 | | | 7.850 | 35.193 |
| 1017 | 47.7 | | | 6.754 | 35.121 |
| 1216 | 41.3 | | | 5.782 | 35.108 |
| *Endeavor cruise 328, Station 4, 22°N 36°E, Sept. 3 1999* | | | | | |
| 0.5 | 28.8 | | | 26.500 | 37.430 |
| 56 | 35.0 | 1.1793 | 2.4469 | 24.292 | 37.463 |
| 102 | 34.9 | 1.1795 | 2.4478 | 22.404 | 37.386 |
| 151 | 39.5 | 1.1784 | 2.4456 | 21.510 | 37.276 |
| 201 | 38.5 | 1.1812 | 2.4461 | 19.852 | 36.923 |
| 296 | 48.5 | 1.1847 | 2.4460 | 17.198 | 36.412 |
| 438 | 65.0 | 1.1881 | 2.4484 | 14.096 | 35.931 |



| | | | | | |
|---|---|---|---|---|---|
| 584 | 89.4 | 1.1880 | 2.4481 | 11.889 | 35.618 |
| 664 | 94.4 | 1.1872 | 2.4478 | 10.563 | 35.456 |
| 765 | 84.6 | | | 9.310 | 35.299 |
| 957 | 49.5 | 1.1847 | 2.4485 | 7.009 | 35.087 |
| 1222 | 41.7 | 1.1852 | 2.4527 | 5.556 | 35.057 |
| 1244 | 36.2 | | | 5.559 | 35.058 |
| 1473 | 25.4 | 1.1872 | 2.4582 | 4.812 | 35.071 |
| 1886 | 24.4 | 1.1859 | 2.4581 | 3.774 | 35.027 |
| 2117 | 18.8 | 1.1873 | 2.4614 | 3.409 | 35.000 |
| 2442 | 13.1 | 1.1881 | 2.4617 | 2.929 | 34.965 |
| 2848 | 17.7 | 1.1899 | 2.4599 | 2.569 | 34.938 |
| 3396 | 14.1 | 1.1910 | 2.4654 | 2.241 | 34.911 |
| 3858 | 9.5 | | | 2.407 | 34.894 |
| 4472 | 13.6 | | | 2.556 | 34.928 |
| 5293 | 8.9 | | | 2.406 | 34.876 |
| *Endeavor cruise 328, Station 5, 26.33°N 33.67°E, Sept. 6 1999* | | | | | |
| 0.5 | 37.8 | | | 26.300 | 37.560 |
| 50 | 39.5 | | | 26.234 | 0.000 |
| 100 | 38.6 | | | 20.685 | 0.000 |
| 150 | 43.6 | | | 19.399 | 0.000 |
| 200 | 44.3 | | | 17.815 | 0.000 |
| 304 | 49.1 | | | 16.130 | 0.000 |
| 439 | 61.4 | | | 13.939 | 0.000 |
| 585 | 86.6 | | | 12.386 | 35.684 |
| 680 | 97.9 | | | 10.794 | 35.505 |
| 781 | 87.8 | | | 9.629 | 0.000 |
| 925 | 71.6 | | | 8.158 | 0.000 |
| 1054 | 64.5 | | | 7.118 | 0.000 |
| 1283 | 42.9 | | | 6.242 | 0.000 |
| 1459 | 44.4 | | | 5.623 | 35.207 |
| 1955 | 34.4 | | | 4.184 | 35.082 |
| 2299 | 25.2 | | | 3.482 | 35.008 |
| 2638 | 20.3 | | | 3.013 | 34.962 |
| 3810 | 14.4 | | | 2.453 | 34.899 |
| *Endeavor cruise 328, Station 6, 27.5°N 29.33°E, Sept. 7, 1999* | | | | | |
| 0.5 | 40.5 | | | 25.600 | 37.440 |
| 52 | 43.9 | | | 25.353 | 37.274 |
| 100 | 45.7 | | | 20.192 | 36.973 |
| 149 | 43.8 | | | 19.566 | 36.972 |
| 198 | 48.5 | | | 18.041 | 36.640 |
| 295 | 53.0 | | | 15.842 | 36.192 |
| 437 | 66.5 | | | 13.819 | 35.880 |
| 590 | 95.0 | | | 10.609 | 35.438 |





| | | | | | |
|---|---|---|---|---|---|
| 682 | 98.6 | | | | |
| 780 | 91.8 | | | 9.604 | 35.397 |
| 933 | 75.5 | | | 8.225 | 35.293 |
| 1078 | 59.4 | | | 7.464 | 35.248 |
| 1272 | 52.0 | | | 6.447 | 35.264 |
| *Endeavor cruise 328, Station 7, 31°N 31°E, Sept. 9, 1999* | | | | | |
| 0.5 | 37.8 | | | 26.000 | 37.050 |
| 47 | 38.8 | 1.1783 | 2.4452 | 23.523 | 36.932 |
| 98 | 32.3 | 1.1783 | 2.4458 | 20.337 | 36.763 |
| 98 | 34.1 | 1.1783 | 2.4445 | 20.337 | 36.763 |
| 148 | 42.2 | | | 7.620 | 34.411 |
| 197 | 41.3 | 1.1829 | 2.4455 | 18.186 | 36.558 |
| 295 | 46.4 | 1.1842 | 2.4471 | 16.676 | 36.294 |
| 436 | 56.8 | 1.1866 | 2.4483 | 14.879 | 36.018 |
| 586 | 58.3 | 1.1888 | 2.4493 | 12.734 | 35.716 |
| 680 | 68.1 | 1.1865 | 2.4468 | 11.259 | 35.520 |
| 782 | 82.4 | 1.1867 | 2.4478 | 9.903 | 35.405 |
| 937 | 75.9 | 1.1846 | 2.4492 | 8.310 | 35.351 |
| 1261 | 62.6 | 1.1821 | 2.4507 | 6.498 | 35.322 |
| 1458 | 65.1 | 1.1812 | 2.4505 | 5.725 | 35.264 |
| 1745 | 49.7 | 1.1806 | 2.4501 | 4.606 | 35.142 |
| 2038 | 43.9 | 1.1803 | 2.4506 | 3.857 | 35.060 |
| 2337 | 36.1 | 1.1808 | 2.4518 | 3.198 | 34.990 |
| 2681 | 16.3 | | | 2.797 | 34.956 |
| 3528 | | 1.1812 | 2.4532 | 2.261 | 34.912 |
| 4016 | 12.6 | | | 2.510 | 0.000 |
| 4311 | 16.5 | | | 2.506 | 0.000 |
| *Endeavor cruise 328, Station 8, 35°N 29°E, Sept. 11 1999* | | | | | |
| 0.5 | 32.8 | | | 25.400 | 36.390 |
| 40 | 40.8 | | | | |
| 94 | 39.8 | | | | |
| 146 | 44.1 | | | | |
| 195 | 45.2 | | | | |
| 294 | 50.3 | | | | |
| 390 | 59.1 | | | | |
| 485 | 66.7 | | | | |
| 586 | 72.9 | | | | |
| 687 | 80.7 | | | | |
| 787 | 80.0 | | | | |
| 856 | 80.6 | | | 9.081 | 35.480 |
| 1025 | 75.2 | | | 7.993 | 35.489 |
| 1130 | 71.4 | | | | |
| 1273 | 68.7 | | | 5.891 | 35.241 |



| | | | | | |
|---|---|---|---|---|---|
| 1518 | 67.0 | | | 4.847 | 35.107 |
| 1764 | 60.5 | | | 3.973 | 34.989 |
| 1960 | 52.5 | | | 3.706 | 34.979 |
| 2152 | 44.9 | | | 3.466 | 34.965 |
| 2352 | 40.0 | | | 3.347 | 34.967 |
| 2943 | 25.2 | | | 2.902 | 34.940 |
| 3091 | 21.1 | | | 2.825 | 34.935 |
| 3242 | 19.8 | | | 2.807 | 34.933 |
| *Endeavor cruise 328, Station 9, 45.52°N 21.48°E, Sept. 15, 1999* | | | | | |
| 0.5 | 40.1 | | | 18.400 | 35.730 |
| 48 | 39.0 | 1.1825 | 2.4481 | 18.756 | 35.750 |
| 146 | 45.9 | 1.1834 | 2.4469 | 13.690 | 35.743 |
| 195 | 50.7 | 1.1839 | 2.4465 | 13.511 | 35.757 |
| 291 | | 1.1871 | 2.4485 | 12.676 | 35.640 |
| 392 | 57.7 | 1.1870 | 2.4482 | 11.909 | 35.545 |
| 446 | 56.3 | 1.1863 | 2.4482 | 11.697 | 35.550 |
| 616 | 69.3 | | | | |
| 641 | 78.9 | 1.1843 | 2.4484 | 9.872 | 35.325 |
| 660 | 82.3 | 1.1842 | 2.4482 | 9.474 | 35.276 |
| 841 | 63.9 | 1.1861 | 2.4506 | 7.508 | 35.156 |
| 1005 | 59.3 | 1.1854 | 2.4510 | 6.281 | 35.140 |
| 1189 | 66.6 | 1.1851 | 2.4514 | 5.068 | 35.038 |
| 1353 | 65.3 | 1.1839 | 2.4504 | 4.264 | 34.961 |
| 1732 | 62.0 | 1.1834 | 2.4489 | 3.571 | 34.899 |
| 2061 | 52.2 | 1.1827 | 2.4482 | 3.333 | 34.896 |
| 2321 | 45.8 | 1.1822 | 2.4490 | 3.282 | 34.922 |
| 2702 | 32.2 | | | 3.050 | 34.942 |
| 2817 | 38.5 | | | 2.958 | 34.943 |
| 2840 | 25.6 | 1.1835 | 2.4507 | 2.944 | 34.943 |
| 3310 | 16.3 | 1.1831 | 2.4520 | 2.727 | 34.929 |
| *Endeavor cruise 328, Station 10, 42°N 17.75°E, Sept. 16 1999* | | | | | |
| 0.5 | 53.5 | | | 20.000 | 35.900 |
| 39 | 50.4 | 1.1797 | 2.4438 | 18.929 | 35.855 |
| 95 | 51.9 | | | 13.555 | 35.762 |
| 147 | 56.0 | 1.1794 | 2.4438 | 13.025 | 35.740 |
| 197 | 54.1 | 1.1808 | 2.4443 | 12.589 | 35.684 |
| 294 | 54.4 | 1.1830 | 2.4468 | 12.014 | 35.612 |
| 441 | 67.0 | 1.1811 | 2.4458 | 11.464 | 35.575 |
| 588 | 76.3 | 1.1830 | 2.4466 | 10.641 | 35.474 |
| 688 | 85.2 | 1.1814 | 2.4458 | 10.464 | 35.536 |
| 780 | 91.3 | 1.1821 | 2.4462 | 10.459 | 35.655 |
| 931 | 90.0 | 1.1804 | 2.4483 | 10.430 | 35.873 |
| 1078 | 88.3 | 1.1800 | 2.4479 | 9.901 | 35.898 |



| | | | | | |
|---|---|---|---|---|---|
| 1272 | 78.8 | 1.1814 | 2.4487 | 7.886 | 35.602 |
| 1440 | 74.2 | 1.1813 | 2.4471 | 5.519 | 35.211 |
| 1680 | 69.5 | 1.1814 | 2.4465 | 4.043 | 34.995 |
| 1864 | 67.5 | | | | |
| 1906 | 77.5 | 1.1819 | 2.4477 | 3.494 | 34.935 |
| 2215 | 45.3 | 1.1796 | 2.4467 | 3.348 | 34.963 |
| 2518 | 35.9 | 1.1788 | 2.4455 | 2.983 | 34.961 |
| 2974 | 33.6 | | | | |
| 3604 | 16.9 | 1.1811 | 2.4508 | 2.281 | 34.915 |
| 4086 | 17.6 | 1.1815 | 2.4522 | 2.176 | 34.905 |

*Endeavor cruise 328, Station 11, 38.58°N 22.28°E, Sept. 19 1999*

| | | | |
|---|---|---|---|
| 0.5 | 52.3 | 22.600 | 36.480 |
| 50 | 50.2 | | |
| 99 | 48.4 | | |
| 150 | 95.9 | | |
| 196 | 63.7 | 14.829 | 35.997 |
| 296 | 67.1 | 13.613 | 35.838 |
| 429 | 72.8 | 12.393 | 35.688 |
| 586 | 91.3 | 11.207 | 35.552 |
| 659 | 81.7 | | |
| 784 | 95.9 | 10.467 | 35.628 |
| 876 | 82.4 | 9.057 | 35.450 |
| 1150 | 80.9 | | |
| 1270 | 77.4 | 7.149 | 35.442 |
| 1635 | 75.7 | 4.690 | 35.088 |
| 1925 | 75.7 | | |
| 2046 | 62.6 | 3.707 | 34.976 |
| 2344 | 51.5 | 3.337 | 34.967 |
| 2557 | 37.8 | 3.128 | 34.958 |
| 2845 | 42.4 | 2.872 | 34.947 |
| 3025 | 31.1 | 2.767 | 34.937 |
| 3331 | 28.3 | 2.652 | 34.928 |
| 3468 | 28.2 | 2.622 | 34.923 |
| 3907 | 30.3 | 2.586 | 34.913 |
| 4212 | 25.1 | 2.568 | 34.908 |





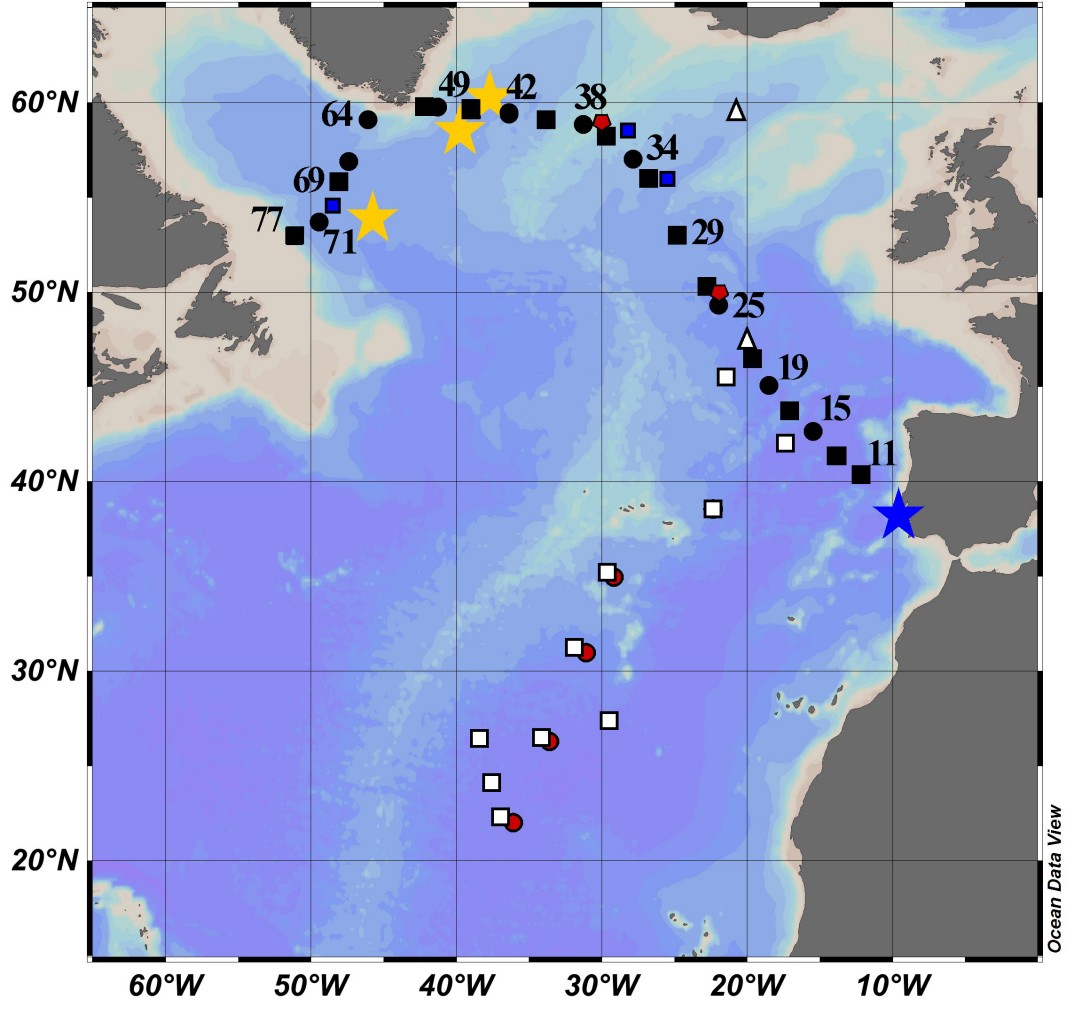

**Figure 1: Map of the cruise transect. GEOVIDE samples are solid black squares (concentration and isotope data) and**
5  **circles (concentration data only). The blue star is GA 03 (2010); the red circles are Atlantis II 123 (1989); the white**
**squares are EN328 (1999); the white triangles are JGOFS (1989); the red pentagons are TTO 1981; the blue squares are**
**IOC-2 (1993); the yellow stars are GA 02 (2010).**





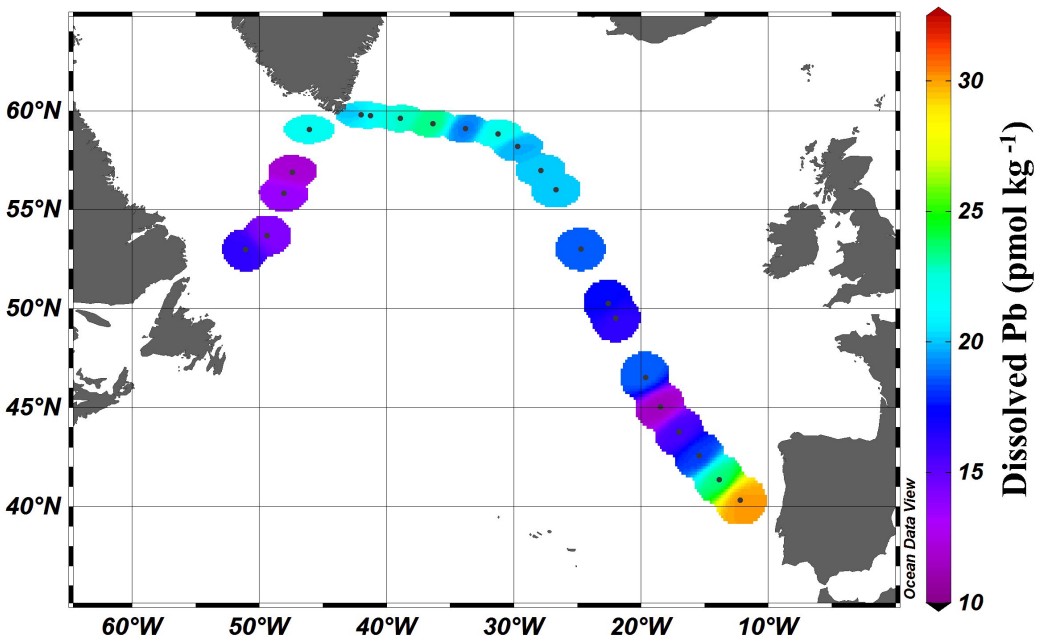

Figure 2: Near-surface (11-20m) concentrations of Pb. Plot created in Ocean Data View (Schlitzer, 2017).

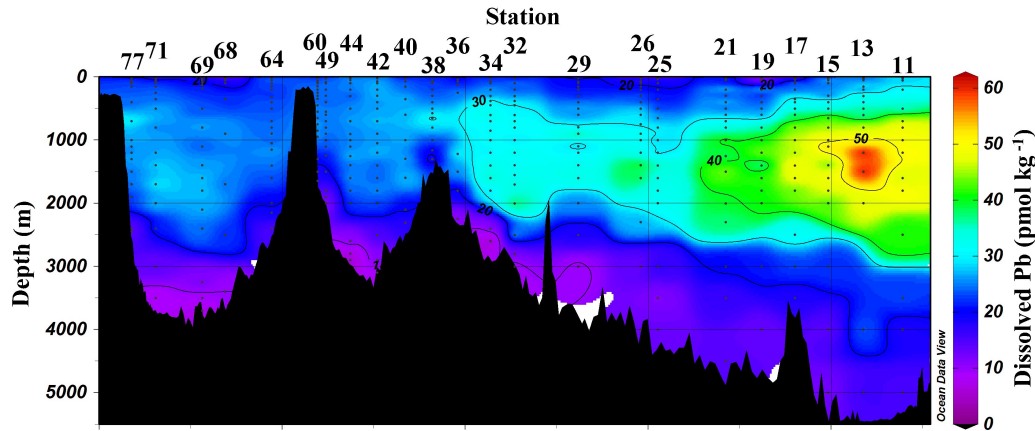

5    Figure 3: Section plot of Pb concentrations in the GEOVIDE section. Plot created in Ocean Data View (Schlitzer, 2017).



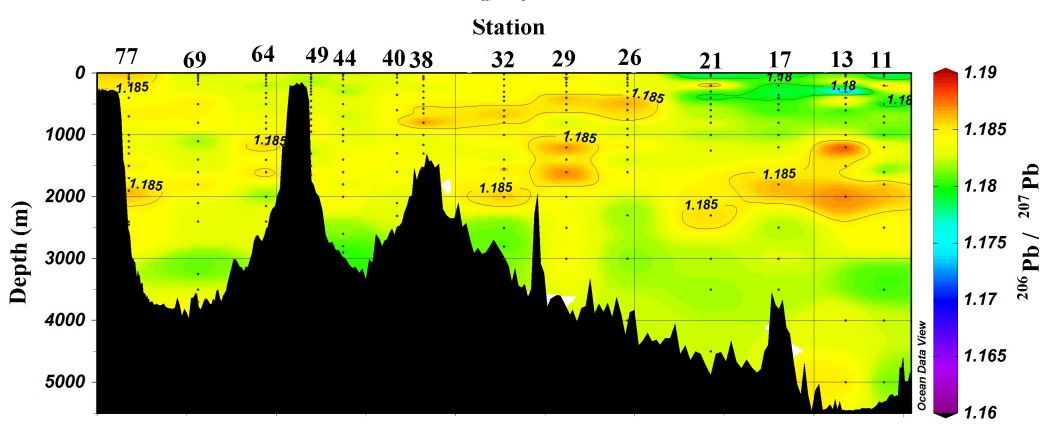

**Figure 4: Section plot of $^{206}$Pb/$^{207}$Pb concentrations in the GEOVIDE section. Plot created in Ocean Data View (Schlitzer, 2017).**



**Figure 5: Pb concentration depth profiles. References: GA03 (Noble et al., 2015); EN328 (this work); JGOFS (Martin et al., 1993); TTO (Weiss et al., 2003); IOC-2 (Veron et al., 1999); GA02 (The GEOTRACES Group, 2015).**



**Figure 6: $^{206}$Pb/$^{207}$Pb isotope ratio depth profiles. References: GA03 (Noble et al., 2015); EN328 (this work); TTO (Weiss et al., 2003); IOC-2 (Veron et al., 1999).**





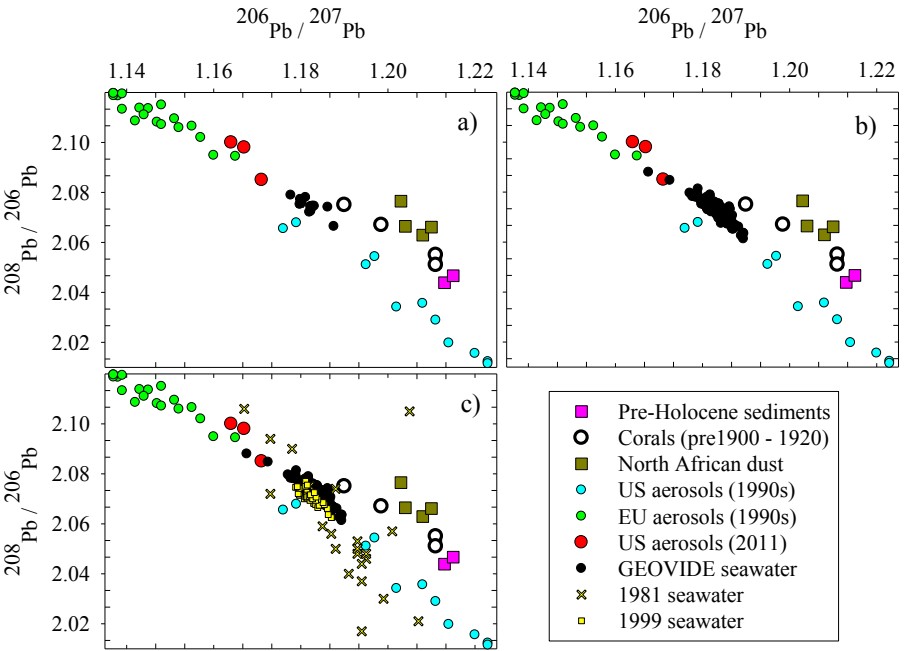

**Figure 7: Triple isotope plot of (a) the surface GEOVIDE samples compared to possible sources, (b) all GEOVIDE data, and (c) as compared with historical seawater data. References: pre-Holocene sediments (Hamelin et al., 1990); corals (Kelley et al., 2009); North African dust (Bridgestock et al., 2016); US and EU aerosols, 1990s (Bollhofer and Rosman, 2001); US aerosols, 2011 (Noble et al 2015); 1981 seawater (Weiss et al., 2003); 1999 and GEOVIDE seawater (this work).**





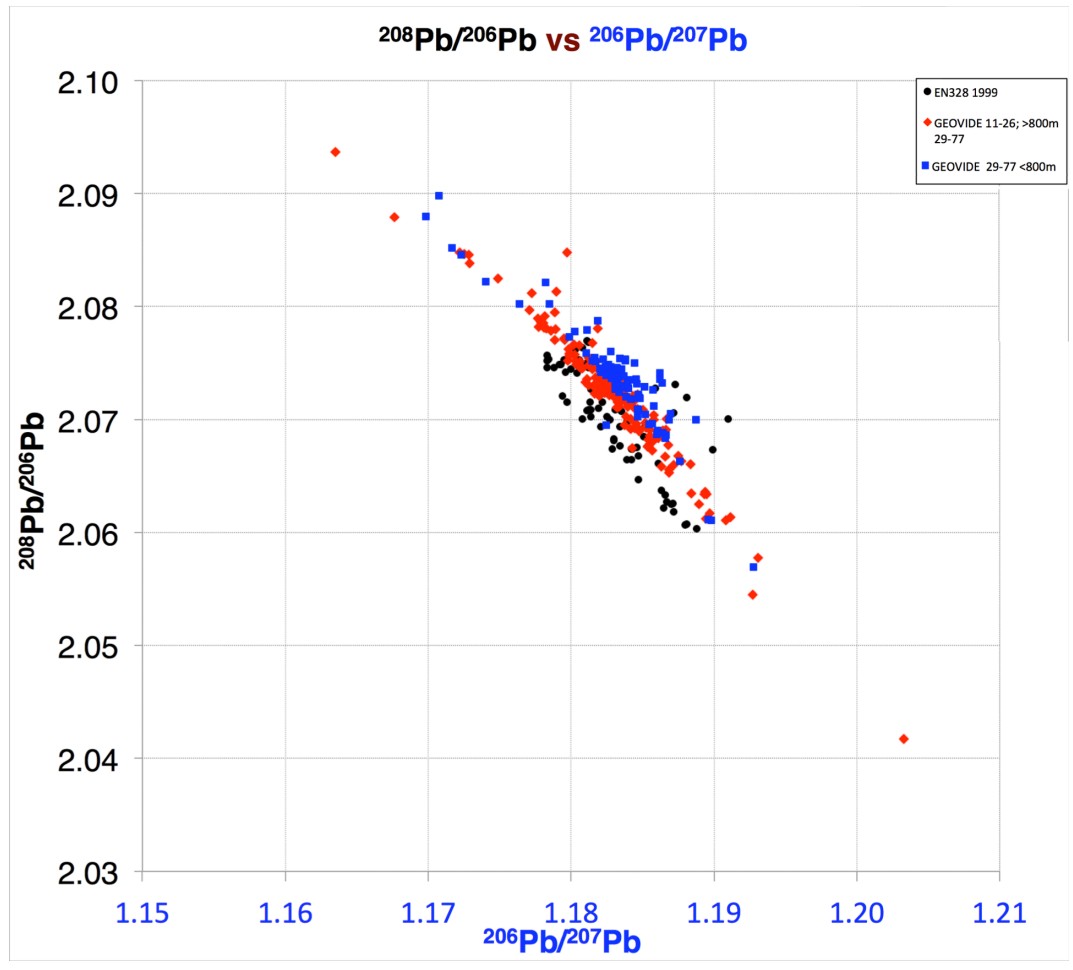

Figure 8: Triple isotope plot of northern North Atlantic Pb in 1999 (EN328, black circles) and 2014 (GEOVIDE, red diamonds: all sample depths from stations 11-26 and samples >800m from stations 29-77; and blue squares: samples <800m from stations 29-77).



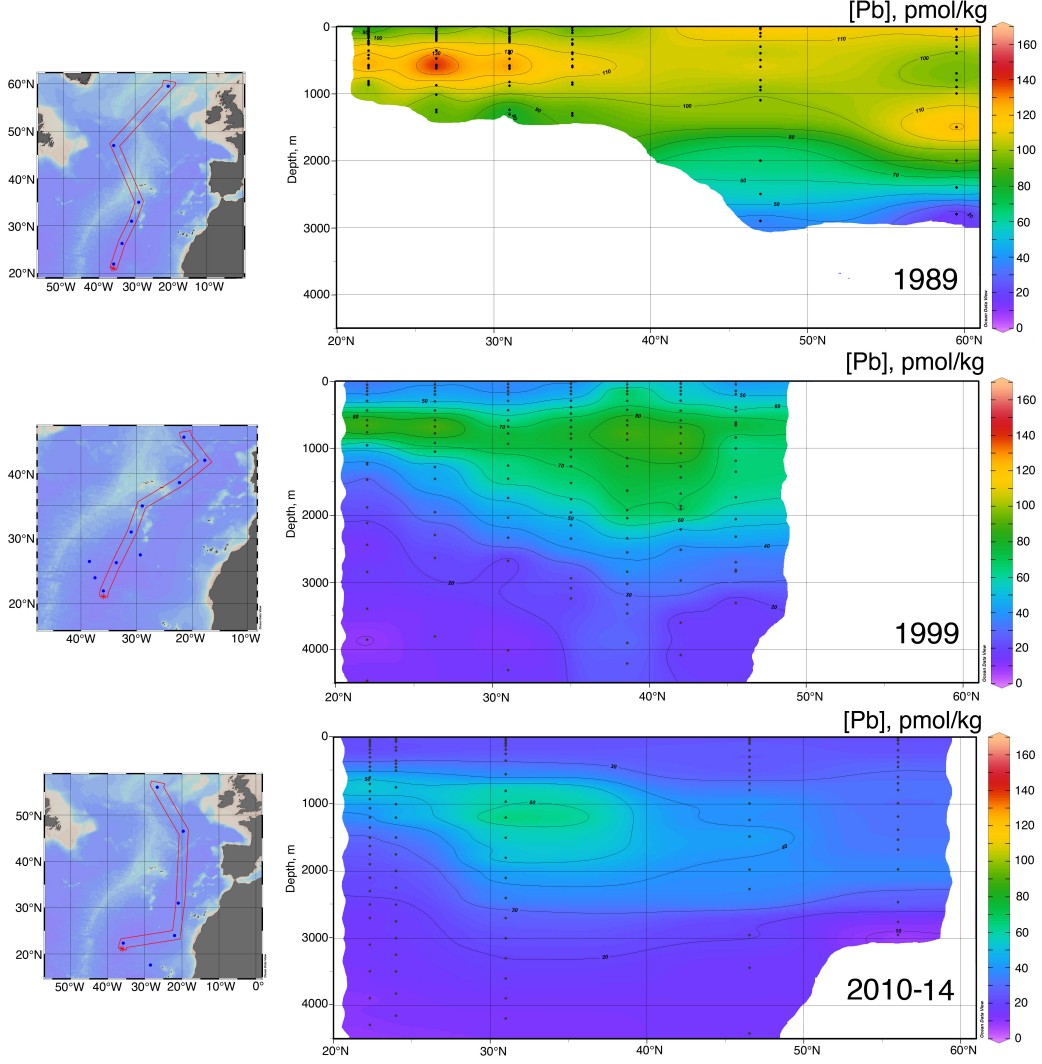

**Figure 9: North-South [Pb] sections in the eastern Atlantic Ocean, 1989-2014. Plot created in Ocean Data View (Schlitzer, 2017).**



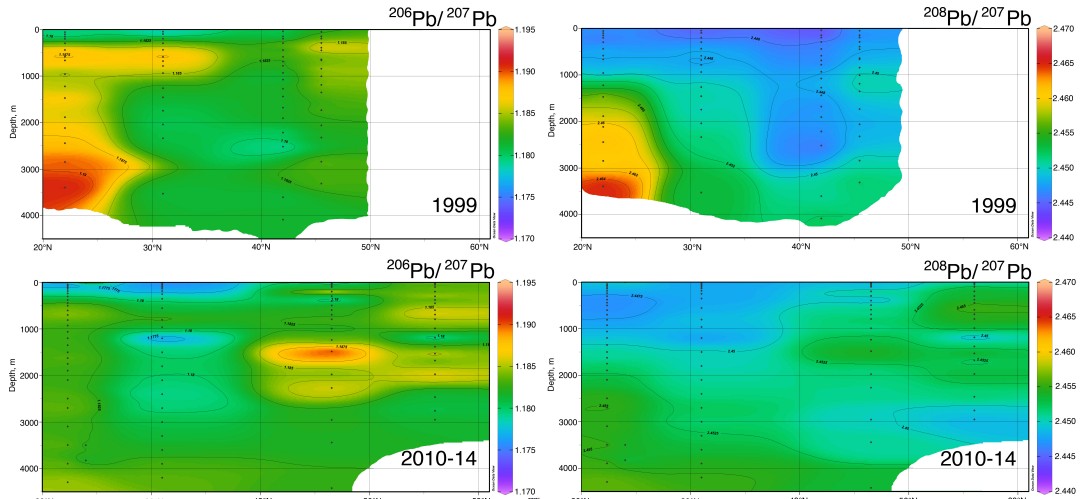

**Figure 10: North-South Pb isotope sections in the eastern Atlantic Ocean, 1999 and 2010-2014. Plot created in Ocean Data View (Schlitzer, 2017).**

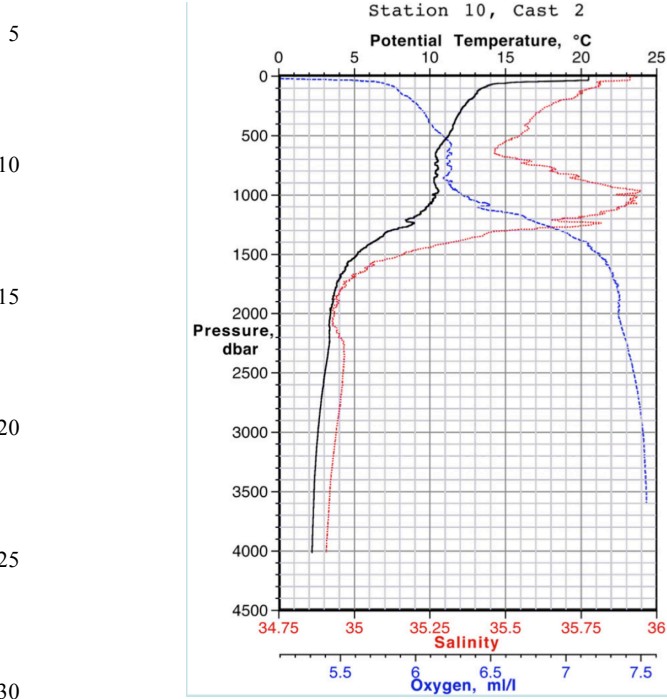

**Figure 11: CTD data from EN328 Station 10 (42°N, 17°45'W) showing strong salinity maximum due to Meddy.**