# Peer review of "Dissolved Pb and Pb isotopes in the North Atlantic from the GEOVIDE transect (GEOTRACES GA-01) and their decadal evolution"

_Biogeosciences, 2018_

## Referee Comment (RC1) · Anonymous Referee #1 · 19 Feb 2018

General comments

Zurbrick et al present an impressively large dataset of dissolved Pb concentrations and isotope compositions from the North Atlantic, spanning both a large geographic region and documenting temporal changes over the past few decades. This allows novel insights into how the sources of this pollutant have changed over time, which will be of interest to a broad group of oceanographers and geochemists alike. The paper is well written and the interpretations and conclusions are generally well supported by the dataset, I therefore recommend its publication in Biogeosciences.

I have just a few recommendations for minor revisions.

[Figure]

One of the main conclusions is that, as of 2014, there is evidence for natural Pb sources to surface seawaters of this region (page 12, lines 22-25). This is based on both the isotopic results of this study and the results of aerosol samples collected during the same cruise presented by Shelley et al. (2017, cited in article). Specifically, Shelley et al., (2017) attributed 60% of the Pb in the atmosphere to be associated with natural mineral dust, and 40% to be from anthropogenic sources (page 10, lines 24-25). The authors then suggest that this ratio of these Pb sources is observed in the dissolved phase of their surface seawater samples (page 10, lines 28-30).

A problem with this line of logic is that it fails to account for the solubility difference between these two atmospheric Pb sources. Lead from anthropogenic emissions is considered to be far more soluble that that associated with mineral dust. This means that deposition of mineral dust derived Pb needs to far exceed that of anthropogenic Pb to produce a ~1:1 ratio of these Pb sources in the dissolved phase of surface seawaters. In other words the roughly 1:1 ratios of natural to anthropogenic Pb determined in aerosols is expected to be substantially modified in surface waters due to solubility differences, resulting in predominantly anthropogenic Pb occurring in the surface waters. This discrepancy needs to be addressed to support the conclusion that naturally sourced Pb is now prominent in surface waters of this region.

Specific comments

Page 4, line 33; What statistic is the '~200ppm' reproducibility of the Pb isotope ratios based on, 2sd? Likewise for the quoted '1000ppm' and '~500ppm' reproducibilites quoted in lines 34 and 36.

Page 6, line 11; what is this 'moderate range in [Pb]? It would be helpful to include specific values here.

Page 8, lines 12-15; both of these cruises have detailed Optimum Multi-Parameter water mass analyses so presumably this interpretation can be verified.

[Figure]

Technical corrections

Page 3, line 19; specify 'samples were analysed for Pb concentrations'

Page 6, line 21; 'Schepanski et al., 2009' is underlined

Page 8, line 31; change 'heavier' to 'higher'

Sections 3.5 and 3.6; references to the appropriate figures become rather sparse in these sections

---

## Referee Comment (RC2) · Anonymous Referee #2 · 26 Mar 2018

This manuscript reports the concentration and isotopic composition data of Pb from the GEOVIDE transect in the North Atlantic and discusses their spatial distribution and decadal change. The quality of the data and discussion is high enough to be published in Biogeosciences. My comments and questions are as follows:

Are the authors using the concentration and isotopic ratio of Pb as conservative tracers in a part of discussion (ex. p. 8, the last paragraph)? Is it valid?

It is interesting that Pb isotope ratios are relatively homogenous and largely decoupled from Pb concentration. I would like to know more details of the mechanism.

In a previous paper (Wu et al., 2010), the authors proposed the pre-industrial

206Pb/207Pb is ~1.210 (based on sediment values) and homogenous in the Pacific deep water. Do you think the Pb isotope ratio in the Atlantic deep water will approach this value because of decrease in anthropogenic effects?

How much is the isotopic fractionation during scavenging and sedimentation? Is it un-detectable? Is it reasonable to assume that the Pb isotope ratio is equal between dissolved species in seawater and fixed species in sediments?

---

## Referee Comment (RC3) · D. Weiss (Referee) · 8 May 2018

This paper by Zubrick and co workers describes a study of Pb and Pb isotopes in the North Atlantic from the GEOVIDE transect and their decadal evolution.

The results confirm recent findings within the community regarding the sources of Pb in the NAO in a post gasoline world, i.e. the suggestion that "natural Pb is coming back" and the high concentrations in subsurface Mediterranean water near the coast of Portugal. There is also an important observation re the homogenous isotopic composition of NA seawater.

[Figure]

The analytical data is of good quality. The data set is extensive and critical to the field in that it allows to assess the temporal evolution of lead in the marine environment since the out phasing of leaded gasoline. It is also a great joy to see how the long-standing efforts of the MIT group and their collaborators to study the marine lead cycle enables us to get unprecedented insights into the global geochemical cycle of lead. This work is as such invaluable and unique and instrumental to push the boundaries of marine chemistry and global geochemical cycles.

I have no hesitation to recommend the paper for publication but would like to add a few reflections that the authors might wish to consider.

First, it would be helpful and good to formulate a proper hypothesis and describe better the aim and objectives of the study. At the moment the authors state that the study evaluates current sources and relative quantities (not sure if that is correct as you determine relative contributions but not quantities as a quantity is defined as amount or number of a material) but I think it would be helpful to be more hypothesis driven and test a specific process or mechanism.

Second, I can understand that the authors want to discuss outliers given the amount of work that goes into getting samples and data – nevertheless, I do wonder what the contribution/value is to publish that 'negative' data. The careful assessment of the data done by the authors suggests that there are contamination issues. If so – why publish? Is the idea of such an assessment not to identify the problems and then report the valid and acceptable data? I have no strong feelings, but I think that chapter (3.1 Outliers) does not add to the paper. If the authors want to keep that chapter, then maybe it would be beneficial for the reader to make clearer what we have learnt from it and how we can prevent it in future.

Third, if feel the authors do not really push the source assessment to the point they could. There is important recent isotope data out on key 'new' potential sources of lead in the atmosphere such as coal, non-combustion vehicle exhausts, diesel etc for

North America and Europe. (Various papers published in EST). I think that data should be added to the source discussion. To this end, it would also be good to be more clear when talking about emissions from what segment they come. I don't think that data is presented or discussed. To this end, I am not sure if the statement re the return of natural Pb is so clear cut. Various studies have shown that coal, tires and brake abrasion etc play a key role as novel source of Pb in the (urban) atmosphere and as the Pb enrichments are still significant in the NA atmosphere, I feel that this needs to be more critically discussed. It is clear to me that this discussion is very difficult and there are many arguments for one or the other, however, I think modern new anthropogenic sources have not been so well included. One question to me seems – how do we reconcile the 10 fold enrichment in the atmosphere with 30 to 50 % natural Pb in the surface waters? I am aware Pb concentrations have come down . . . But are they half way back to 'normal'? Do we have a number for pre-anthropogenic Pb concentration?

Forth, I am not sure how much mineral dust is really a source of natural lead given the very low solubility of silicates. With respect to present days, we know that anthropogenic particles are much more soluble (see various recent Nature Geoscience papers) and hence could possibly control the dissolved Pb budget even with a small enrichment. With respect to pre-anthropogenic times, I wonder if we can ignore the importance of passive volcanic degassing. A series of papers have shown that passive degassing of volcanoes can be a very important source of trace metals to the atmosphere and given that in this case Pb is either in gas phase or a more soluble silicate phase, that could be an important source too.

Finally, a more editorial point. I think the amount of figures can be reduced

I hope my comments are helpful and wish the authors only the best. A very fine contribution.

---

## Author Comment (AC1) · 19 Jun 2018

Reply to reviewers' specific comments (within quotes) (their line numbers refer to original manuscript) (our response in plain text, line numbers refer to revised version)

Reviewer 1:

"In other words the roughly 1:1 ratios of natural to anthropogenic Pb determined in aerosols is expected to be substantially modified in surface waters due to solubility differences, resulting in predominantly anthropogenic Pb occurring in the surface waters. This discrepancy needs to be addressed to support the conclusion that naturally

sourced Pb is now prominent in surface waters of this region."

We have addressed this issue specifically in several places using the interpretation of the reviewer, most in detail in the section beginning with p. 11, line 33 to p. 12, line 7. Specific comments

"Page 4, line 33; What statistic is the '200ppm' reproducibility of the Pb isotope ratios based on, 2sd? Likewise for the quoted '1000ppm' and '500ppm' reproducibilites quoted in lines 34 and 36."

Because the isotope ratio precision is not constant for all of these samples (mainly because of fixed sample size with Pb concentrations varying over an order of magnitude), this statement is a qualitative assessment based on our examination of replicates over the entire data set. To offset this subjectivity, we have included some specific statistics for the pooled standard deviation of duplicates for specific concentration ranges from the data shown in supplement figure S2 (new table 1)."

"Page 6, line 11; what is this 'moderate range in [Pb]? It would be helpful to include specific values here."

Done as requested.

"Page 8, lines 12-15; both of these cruises have detailed Optimum Multi-Parameter water mass analyses so presumably this interpretation can be verified."

eOMP discussion added : , p. 8 lines 36-37, p.9 lines 26-30

"Technical corrections

Page 3, line 19; specify 'samples were analysed for Pb concentrations' Page 6, line 21; 'Schepanski et al., 2009' is underlined Page 8, line 31; change 'heavier' to 'higher' Sections 3.5 and 3.6; references to the appropriate figures become rather sparse in these sections"

All of these are addressed in the revised version.

---

## Author Comment (AC2) · 19 Jun 2018

Reply to reviewers' specific comments (in quotes, their line numbers refer to original manuscript); our response in plain text, line numbers refer to revised version) Reviewer 2:

"Are the authors using the concentration and isotopic ratio of Pb as conservative tracers in a part of discussion (ex. p. 8, the last paragraph)? Is it valid?"

We don't claim anywhere that Pb is a conservative tracer. However, the signatures imparted at the surface are advected into the interior and hence deep water masses

do reflect their sources to some extent. The Pb isotope ratio is less influenced by scavenging (little change despite Pb removal).

"It is interesting that Pb isotope ratios are relatively homogenous and largely decoupled from Pb concentration. I would like to know more details of the mechanism."

Although Pb sources have varied by an order of magnitude over the past half century, the isotope ratios of the sources have varied less (e.g., Europe always has lower 206Pb/207Pb than North America). To the extent that European and North American emission sources have varied in tandem, the isotope ratios of the ocean don't change very much. This is not strictly true and we acknowledge this in our other papers, e.g. (see reference Kelly et al. 2009: both North American and European Pb sources have evolved over time, and slight differences in the timing of North American and European Pb gas phaseout have altered the proportions seen in the Atlantic Ocean from each source.

"In a previous paper (Wu et al., 2010), the authors proposed the pre-industrial 206Pb/207Pb is 1.210 (based on sediment values) and homogenous in the Pacific deep water. Do you think the Pb isotope ratio in the Atlantic deep water will approach this value because of decrease in anthropogenic effects?"

Wu et al. (2010) were referring to the deep Pacific basin which does not have advected anthropogenic lead, instead transported by sinking particle exchange (in contrast to the advectively dominated Atlantic "bowling alley". But yes, our coral work (Kelly et al., 2009) show that two centuries ago, the Atlantic Pb isotope ratio was similar to typical crustal materials. In time (many decades to a century), the Atlantic could revert to this value if all anthropogenic emission sources were eliminated. But anthropogenic emissions to this basin are still significant despite the elimination of leaded gasoline from automobiles.

"How much is the isotopic fractionation during scavenging and sedimentation? Is it un-detectable? Is it reasonable to assume that the Pb isotope ratio is equal between

dissolved species in seawater and fixed species in sediments?"

First, it is impossible to determine whether Pb undergoes significant isotope fractionation in the environment because there is no non-radiogenic isotope pair to evaluate this (unlike for Nd or Sr). It could be done in the laboratory under controlled situations but I am not aware of any experiments that demonstrate fractionations. Second, it isn't implausible that small isotope fractionations can occur for Pb. After all, they are seen for some other heavy isotope ratios (e.g. Tl, Hg, U). But these fractionations are never more than a few per mil – which is much smaller than the >15% range known from radiogenic signatures. Such small variations are close to our analytical precision and could never be reliably demonstrated.

---

## Author Comment (AC3) · 19 Jun 2018

Reply to reviewers' specific comments (in quotes, their line numbers refer to original manuscript); our response in plain text, line numbers refer to revised version)

Reviewer 3:

"First, it would be helpful and good to formulate a proper hypothesis and describe better the aim and objectives of the study. At the moment the authors state that the study evaluates current sources and relative quantities (not sure if that is correct as you determine relative contributions but not quantities as a quantity is defined as amount or

number of a material) but I think it would be helpful to be more hypothesis driven and test a specific process or mechanism."

Changed as per lines 30-35 in revised version.

"Is the idea of such an assessment not to identify the problems and then report the valid and acceptable data? I have no strong feelings, but I think that chapter (3.1 Outliers) does not add to the paper. If the authors want to keep that chapter, then maybe it would be beneficial for the reader to make clearer what we have learnt from it and how we can prevent it in future."

We respectfully disagree with this perspective. First, even though much has been learned much about how to control Pb contamination during seawater sampling, this understanding isn't very well diffused throughout the ocean geochemical community, even amongst trace element analysts. If we have been asked at the outset, we would have recommended extensive testing of the sampling system for Pb before the cruise. But this wasn't done (probably because the sampling system wasn't delivered until just before the cruise). Yet we want people to understand that they have to look at their Pb data critically and not assume that just because some of it makes sense, it all must be correct. Second, there are examples in the literature of people leaving out data that they thought were influenced by contamination but turned out to be correct because of a process that the authors weren't aware of (example: the North Atlantic JGOFS Fe data, as later unearthed by Dutch scientists). By publishing this data with a flag that says I don't believe this, it lets other people with more information in the future re-evaluate this conclusion. This way of presenting data is more and more the standard in the GEOTRACES community, through, for example, the Intermediate Data Product.

"I feel the authors do not really push the source assessment to the point they could. There is important recent isotope data out on key 'new' potential sources of lead in the atmosphere such as coal, non-combustion vehicle exhausts, diesel etc for North America and Europe. (Various papers published in EST). . ..it would also be good to be

more clear when talking about emissions from what segment they come.. . .I don't think that data is presented or discussed. To this end, I am not sure if the statement re the return of natural Pb is so clear cut. To this end, it would also be good to be more clear when talking about emissions from what segment they come. I don't think that data is presented or discussed. However, I think modern new anthropogenic sources have not been so well included."

We have made some attempt to track down the EST papers that this comment addresses (I've looked at 35 from the past 20 years). Although in many cases we acknowledge the presence of the particular source (e.g. Pb from automobile wheel balancing weights), we have little knowledge of their isotope ratios or their transport into the atmosphere and ocean. To really address this matter would require a whole paper most of which wouldn't be very relevant to the oceanic dataset at hand. So we mention minor sources but have not carried out an extensive analysis. If the reviewer thinks this deserves a more thorough treatment, perhaps we could get together on some future manuscript.

"One question to me seems – how do we reconcile the 10 fold enrichment in the atmosphere with 30 to 50 % natural Pb in the surface waters? I am aware Pb concentrations have come down : : : But are they half way back to 'normal'? Do we have a number for pre-anthropogenic Pb concentration?"

The ten-fold enhancement number is based on GLOBAL emissions data, and does not apply to all regions simultaneously. So although emissions from Europe and America have dropped a lot in the past 30 years, those from Asia have increased (see discussion in Boyle et al. (2015) Oceanography). We know that near Bermuda, Pb concentrations of surface waters have declined from 160 pmol/kg in 1979 to <20 pmol/kg in 2011. Coral Pb records imply that near-Bermuda Pb was about 200 pmol/kg in the mid-1970's (Kelly et al. 2009), so yes, there has been nearly an order of magnitude drop in Pb in the Atlantic surface waters – about 2/3 of which is attributable to Pb gasoline phaseout and the remaining portion due to other emission controls. Surface coral data implies

that Atlantic surface [Pb] was about 15 pmol/kg in 1780 (Kelly et al. 2009) and deep coral data implies that [Pb] at 1400m depth was about 3-11 pmol/kg pre-1700 (Lee, 2017 EPSL 458:223).

"Fourth, I am not sure how much mineral dust is really a source of natural lead given the very low solubility of silicates."

Yigal Erel (Erel et al. 1991 GCA 55:707 and Erel et al. 1992 GCA 56:4157) has shown that the primary Pb-containing minerals are largely destroyed during weathering and the released Pb is adsorbed onto the surfaces of mineral phases such as iron oxides. So the dissolution of silicate minerals is not required for the release of Pb from mineral dusts. As referenced, Bridgestock et al. has made the case for some detectable presence of crustal Pb in the tropical Atlantic ocean recently, and Chen et al. (2017, MarPollBull 116:469) have shown that crustal Pb is exchanges with dissolved anthropogenic Pb in continental shelf waters.

"With respect to pre-anthropogenic times, I wonder if we can ignore the importance of passive volcanic degassing."

Russ Flegal (Flegal et al. (1993, Nature 365:242) has argued that volcanic Pb can be seen in the Antarctic, and we know that volcanic emissions have high Pb because it is volatile at magmatic temperatures. So yes, it is a possible source, but we can't say anything about whether it influences the present northern North Atlantic Pb.

"Finally, a more editorial point. I think the amount of figures can be reduced."

We have eliminated one figure.